# Fabrication of Hydrophobic Ni Surface by Chemical Etching

**DOI:** 10.3390/ma12213546

**Published:** 2019-10-29

**Authors:** Xiaojing Qian, Tao Tang, Huan Wang, Changan Chen, Junhong Luo, Deli Luo

**Affiliations:** Institute of Materials, China Academy of Engineering Physics, Jiangyou, Mianyang 621907, China; angie903@163.com (X.Q.); tangtao@caep.com (T.T.); wanghuan@caep.com (H.W.); chenchangan@caep.com (C.C.); qianxiaojing@caep.com (J.L.)

**Keywords:** nickel, wettability, hydrophobicity, chemical etching

## Abstract

Hydrophobic surfaces were successfully fabricated on pure nickel substrates by a one-step chemical etching process with different acidic solutions. The static water contact angle (SCA) of the etched Ni surfaces reached higher than 125°, showing excellent hydrophobicity. The examination of surface chemical compositions implied that there were almost no polar moieties on the surface after chemical etching, except part of the surface was oxidized. After chemical etching, the nickel surfaces became much rough with packed terrace-/crater-/thorn-like clusters. According to the analysis of surface composition and morphology, the hydrophobicity was evidently attributed to the rough microstructures on the etched Ni surface. The best hydrophobicity on Ni surface was produced with the SCAs as high as 140.0° by optimizing the etching time and etchants. The results demonstrate that it is possible to construct hydrophobic surfaces on hydrophilic substrates by tailoring the surface microstructure using a simple chemical etching process without any further hydrophobic modifications by low surface energy materials.

## 1. Introduction

Wettability is one of the most important properties of solid surfaces, and depends on both the surface microstructure and the chemical composition [1,2,3]. Thereby, the hydrophobicity of surface can be tuned by tailoring of surface roughness or adjusting of its surface free energy [4,5,6,7,8,9,10]. The latter method has been usually applied to prepare hydrophobic or superhydrophobic surfaces on metal or alloy substrates which are intrinsically hydrophilic. Various methods have been developed, including plasma etching [11,12], anodic oxidization [13,14], chemical vapor deposition [15,16], electrochemical deposition [17,18,19], chemical etching [19,20,21,22,23,24], sol-gel [25,26], laser processing [21,23,27], and so on. Hydrophobic or superhydrophobic surfaces reported have been constructed on metal and alloy substrates, such as Mg [28], Cu [23,27,29], Zn [1,30], Al [1,22,31], and Al alloy [5,32]. However, most of them have to introduce low surface energy materials in order to obtain the hydrophobic or superhydrophobic surface.

Nishino et al. [10] have demonstrated that the maximal static water contact angle on the lowest surface free energy based on -CF_3_ alignment was almost impossible to increase up to 120°. Therefore the surface roughness plays an important role in forming a hydrophobic or superhydrophobic surface on intrinsically hydrophilic materials [8,33]. Great efforts have been employed to enhance surface roughness. Shirtcliffe et al. [29] investigated the electrodeposited copper layer with varying surface roughness and the SCA on the layer was 115°. Zhang et al. [24] produced a hydrophobic surface on black silicon with the mean SCA of 118° using metal-assisted wet chemical etching. Chen et al. [8] fabricated a superhydrophobic surface on aluminum substrate with the SCA of 154.8° using chemical etching.

Nickel is an excellent catalytic material. According to some catalytic reactions which need hydrophobic Ni surface and do not want to introduce organic matter, it is a great challenge to create a hydrophobic or superhydrophobic surface on pure Ni metal without employing low surface energy materials. Although many works have already published about hydrophobic or superhydrophobic surface of Ni, part of them need to introduce low surface energy materials to enhance the hydrophobicity or obtain the superhydrophobic surface [6], another part is about the hydrophobic Ni films produced on other substrates [17,18,34]. In this work, a hydrophobic surface was fabricated on pure Ni substrate using a one-step chemical etching process without introducing low surface free energy materials. Chemical etching is one of the most applicable ways to construct hydrophobic surfaces on metal substrates due to its simple, lower requirement for equipment. The resulting surfaces were characterized by SCA measurement, energy dispersive X-ray spectroscopy (EDS), X-ray photoelectron spectroscopy (XPS), attenuated total reflection-flourier transformed infrared spectroscopy (ATR-FTIR), and field emission scanning electron microscope (FE-SEM) to evaluate changes in surface wettability, composition, and morphology. The nickel surfaces with the presence of terrace-/crater-/thorn-like rough structure and nonpolar functional groups contributed to excellent hydrophobicity. The chemical etching process is capable of constructing rough microstructures on Ni surface, leading to an enhancement of hydrophobicity.

## 2. Materials and Methods

### 2.1. Materials

Nickel plates (1.5 mm thick; purity ≥99.6 wt%) were purchased from Jinchuan pure metal material Corp. (Jinchang, Gansu, China) and cut into 10 mm × 10 mm pieces for experiments. Chemical reagents such as acetone, sodium hydroxide (NaOH, 99%), hydrochloric acid (HCl, 36%–38%), nitric acid (HNO_3_, 65%–68%), sulfuric acid (H_2_SO_4_, 98%), and phosphoric acid (H_3_PO_4_, 85%) were supplied by Kelong Chemical Reagents Corp. (Chengdu, China). The ethanol (99.7%) and hydrofluoric acid (HF, 40%) were obtained from Guanghua Chemical Reagents Corp. (Shantou, China). All the chemical reagents used in this study were of analytical grade without any further purification. The deionized water was synthesized in the lab.

### 2.2. Samples Preparation

Prior to chemical etching, the Ni plates were firstly ultrasonically washed by successive immersion in acetone and deionized water for about 20 min, and then were ultrasonically degreased with 1 mol/L NaOH aqueous solution for 1 h at 30 °C. Finally, the Ni plates were ultrasonically cleaned with acetone, alcohol and deionized water for 10 min, respectively.

Strong acids such as HCl (10.2 wt%, 15.1 wt%, and 20.0 wt%), H_2_SO_4_ (10.1 wt%, 15.0 wt%, and 20.0 wt%), H_3_PO_4_ (10.0 wt%, 15.0 wt%, and 20.4 wt%), HNO_3_ (10.1 wt%, 15.1 wt%, and 20.0 wt%) and HF with the concentration of 40.0 wt% were chosen as the etchants. The Ni plates were dipped into different acidic solutions at 30 °C and characterized after etching for each one hour. The samples were placed in a digital constant temperature water bath (HH-S1, Jintan, China) to control the experimental temperature before characterization, the samples were ultrasonically washed with alcohol and deionized water for 10 min, respectively, in order to remove any potential surface contaminants. Finally, samples were dried at 40 °C for 30 min.

### 2.3. Characterization

The SCA measurements were performed by an optical contact angle measuring instrument (Krüss DSA30, Hamburg, Germany) using the sessile drop method at ambient temperature with deionized water. The SCA was measured by the image of a drop of 1–2 μL dripped on the sample. The average value of more than five measurements obtained at different positions on the sample surface was adopted as the contact angle.

The chemical composition on the nickel surface was characterized by EDS (EDAX Octane Pro, Mahwah, NJ, USA), incorporated in the FE-SEM and XPS. XPS analysis was performed using an AXIS SUPER spectrometer (Kratos Analytical Inc., Stretford, UK) with a monochromatic Al Kα X-ray source (hν = 1486.6 eV, 150 W). Survey scans were performed at pass energy of 160 eV, whilst narrow scans were performed at pass energy of 20 eV, and the overall energy resolution was 1.2 eV. XPS surface analysis was directly conducted without sputtering after the sample being dried. The base pressure of the instrument was 10^−9^ Torr. XPS spectra were aligned using the C 1s peak at 284.6 eV as a reference and recorded with analysis spot of 60–100 μm in diameter and sampling depth of about 10 nm.

The FTIR was used to evaluate the adsorption on the surface with a high sensitivity to the adsorbate. IR spectra were obtained with Thermo Scientific Nicolet iS50 FTIR spectrometer (Waltham, MA, USA), using ATR technique, in the spectral range of 400–4000 cm^−1^, with a resolution of 2 cm^−1^ accumulating 64 scans. The ATR-FTIR spectra were recorded at room temperature and all spectra were background corrected.

The surface morphology of nickel surfaces was detected by the FE-SEM (FEI Sirion 200, Hillsboro, OR, USA) with the operating voltage of 10 kV and 5 kV. The surface roughness was obtained by Laser roughness measuring instrument (Taylor Hobson, pgi 840, Leicester, UK). The average value of Ra about more than five measurements obtained at different positions on the sample surface was adopted as the surface roughness.

## 3. Results and Discussion

### 3.1. Surface Wettability

The wettability of the nickel surface before and after chemical etching was characterized by measuring the SCAs using an optical contact angle instrument. Figure 1 shows the optical images of water droplets on the Ni surface. Nickel and its alloys are a type of hydrophilic material. The SCA of pure Ni surface before etching is 32.2° (Figure 1a). After immersing in the aqueous solution of 10.2 wt% HCl and 10.1 wt% H_2_SO_4_ for 42 h, the SCAs increase up to 144.5° and 147.2°, respectively (Figure 1b,c). While the SCAs are improved up to 137.7° and 125.1° after etching by the acidic solutions of 10.0 wt% H_3_PO_4_ and 10.1 wt% HNO_3_ for 56 h, respectively (Figure 1d,e). The SCAs quickly rise to 136.6° after etching with 40.0 wt% HF for 28 h (Figure 1f). The etched samples show significant increases in the SCAs compared with the untreated sample. The SCAs increase from 32.2° to at least 125° after the chemical etching process, suggesting that the wettability of Ni surface changes from hydrophilic to hydrophobic. The values of the SCA indicate that the etched surfaces show excellent hydrophobicity.

### 3.2. Surface Chemical Composition

The chemical composition of the untreated Ni substrates was evaluated by an X-ray fluorescence spectrometer. The results indicate that Ni substrates are nearly pure Ni, consisting of 99.6 wt% Ni. Figure 2 shows the elemental compositions as found on the Ni surfaces before and after chemical etching from the EDS analysis. The spectra appear to be similar, the major element found on the surfaces is Ni, accounting for over 97.5% of the surface composition, the appearance of carbon and oxygen can be attributed to the adsorption in air, for these samples are exposed to air prior to EDS analysis.

XPS analysis was conducted to determine the surface chemical composition. Figure 3 shows the XPS survey scans of the Ni surface before and after chemical etching. As described above, all samples contain nickel, carbon, oxygen, and nitrogen elements on the surfaces, this arises from air exposure before XPS analysis [35]. The O 1s binding energy (BE) located at 531.2 eV for all the samples is mostly attributable to adsorbed oxygen. The low peak at BE ≈ 398 eV indicates the presence of N 1s.

The C 1s region of XPS spectra is presented in Figure 4. C1s signal for these samples generates a dominant peak roughly at 284.6 eV and this dominant peak can be attributed to “adventitious C” as a result of carbon adsorption in air. The peak widths are almost unchanged, while the heights of these peaks are different. The peak height of the etched sample is higher than the one of untreated. Based on the C signal value, C intensity levels on the Ni surface increase after chemical etching, suggesting that carbon content in the etched sample is probably higher than the one untreated. The results indicate that the carbon impurities in air are easier adsorbed by the surface after chemical etching.

Figure 5 displays the narrow scans of Ni XPS spectra. The observed BEs from 852 eV to 856 eV indicate the presence of pure Ni, NiO, and Ni_2_O_3_ on the surface even after etching, this observation is consistent with the EDS analysis. The BEs of 2p3/2 peak of pure Ni are within 0.2 eV of 852.2 eV, which agree closely with the value reported in the literature for metallic nickel [36,37,38]. There are no visible variations in the shapes of the Ni 2p3/2 peak, apart from an increase of peak intensity after etching. The comparatively low-intensity of Ni signal is found for the untreated sample. Ni signal intensities increase as reaction with the acidic solutions, when using HNO_3_, the intensity reaches the highest. Ni is formally divalent and bonded to oxygen, the Ni 2p3/2 peaks of NiO are located at about 854 eV span a 0.3 eV range in Figure 5a–d, in accordance with the values recorded in references [36,37,38,39]. While Ni 2p3/2 peaks of Ni_2_O_3_ are within about 0.2 eV of 855.6 eV for all samples, which is consistent with the reference value in 36. A clear shift of Ni 2p3/2 intensities of NiO to a lower value has been observed after etching. Similar behavior has also been observed on the Ni 2p3/2 of Ni_2_O_3_. These shifts result from oxidation state changes to Ni [40,41]. The results indicate that the Ni surface will be gradually covered by NiO and Ni_2_O_3_ with the presence of air. After acidic etching, part of the oxides, especially NiO, is reacted, result in the increase of Ni 2p3/2 peak intensity. As shown in Figure 5, among the acidic solutions used in this work, HCl, HNO_3,_ and HF are effective in removing nickel oxide.

The adsorption on the nickel surfaces before and after chemical etching was carried out by ATR-FTIR. Figure 6 shows the FTIR spectra after background subtraction and baseline corrected. Two low-intensity vibrational bands have appeared in the fingerprint region of the FTIR spectra at around 412 cm^−1^ and 483 cm^−1^. The band appeared at 412 cm^−1^ corresponds to the vibration of Ni-O bond [42]. Another band at 483 cm^−1^ is ascribed to the nickel hydroxide [43]. FTIR spectra at wavenumbers 3900–3600 cm^−1^, 1060 cm^−1^, and 670 cm^−1^ may be assigned to the stretching vibration in the alcohol group [44,45,46,47]. The spectra appeared at 670 cm^−1^ and 1060 cm^−1^ are due to the vibrations of C-H bond [44]. There are three high-intensity bands at around 3610 cm^−1^, 3740 cm^−1^, and 3850 cm^−1^. According to the data from literature [45,46,47,48], these bands are assigned to the -OH vibrations. A low-intensity band at about 3660 cm^−1^ is clearly observed in Figure 6b–f. It has been reported that the band occasionally observed at 3660 cm^−1^, sometimes between 3650 cm^−1^ and 3700 cm^−1^, corresponds to small amount of water adsorbed on the surface [47,49]. The FTIR data obtained on the Ni surfaces show great similarity, and the spectra are hardly affected by chemical etching, except the intensities of the absorption bands assigned to the vibrations of C-H and O-H increase after etching. It is probably that the fresh surface can easily adsorb ethanol and H_2_O molecules after chemical etching. The results indicate that the surfaces of the nickel are oxidized, and the drying process can not completely remove the ethanol and H_2_O molecules after cleaning. The IR spectra are in accord with the XPS spectra. In addition to minimum C-H and O-H group density displayed in Figure 6, there are almost no polar moieties on the surface after chemical etching.

As described above, the chemical composition on the sample surfaces is almost not changed after chemical etching. For the surfaces with the similar chemical composition, differences in surface wettability are expected to be correlated to the surface microstructures, since surface microstructure (i.e., surface roughness) and chemical composition (i.e., surface free energy) are two main factors governing the surface wettability [4,5,6,7,8].

### 3.3. Surface Morphology

The geometrical characteristics of the Ni surfaces before and after chemical etching have been examined under an FE-SEM, the results are shown in Figure 7. As expected, the morphologies of the Ni surfaces before and after chemical etching are obviously changed. It can be seen that the untreated Ni surface (Figure 7a) is much smoother than the etched Ni surfaces except for some scratches caused by the Ni plate manufacturing process. Figure 7 illustrates the significant differences in surface morphologies after chemical etching by different acidic solutions. During the chemical etching process, the components on the surface react with the acids. As a result, there is a great deal of irregular and various etch pits on the surface. After a definite etching time, terrace-like (Figure 7b), crater-like (Figure 7c,d), or thorn-like (Figure 7e,f) microstructures are observed on the surfaces, and the microstructures have different distributions on the etched surfaces.

The values of surface roughness are shown in Table 1. The value of Ra on the untreated sample surface is 1.33 μm. The irregular microstructures formed after etching will enhance the surface roughness. After an etching by 10.2 wt% HCl and 10.1 wt% H_2_SO_4_ for 42 h, the value of Ra increases to 6.24 μm and 6.59 μm, respectively. This unique surface texture contributed to trapping large amounts of air and forming the air cushion underneath the water droplet, which can prevent the liquids contacting the nickel substrate [50,51,52]. So the SCAs increase after etching. As shown in Table 1, the increase of SCAs is accompanied by an increase of roughness. The greater the roughness, the higher the SCA. When etching by 10.1 wt% H_2_SO_4_ for 42 h, Ra increases from 1.33 μm to 6.59 μm, and the SCA increases from 32.2° to 147.2°. Such the surface SCA changed accompanying by the roughness is also observed by using other acidic solutions, as shown in Table 1.

In addition, the surface morphologies are observed to be transformed by extending the etching time, as shown in Figure 7. For instances, Figure 7b shows that the micro-scale terrace structures grow and big etch pits are present after further etching for 14 h using 10.2 wt% HCl, as a result, the surface roughness decreases from 6.24 μm to 4.58 μm, and the SCA on the etched surface decreases from 144.5° to 125.5°. After etching for 49 h by 10.1 wt% H_2_SO_4_, the micro crater-like structures are destroyed (Figure 7c) compared with the surface morphology etching for 42 h. As the surface roughness decreases from 6.59 μm to 5.70 μm, the SCA changes from 147.2° to 135.5°. Such surface morphologies affected by etching time are also observed by using other acidic solutions (Figure 7d–f). The results indicate that the hydrophobic morphologies may be destroyed by excessive etching and the SCAs will decrease as the surface roughness decrease. According to the description of the chemical composition above, the surfaces with a similar chemical composition, the surface roughness is considered to be the primary factor leading to the hydrophobicity. Thus, optimizing the etching conditions, such as etching time, is favorable to obtain the proper morphologies for hydrophobicity.

### 3.4. Effect of Etching Time and Etchants

The influence of the etching time on the hydrophobicity of the etched Ni surface was studied. The relationship between the SCAs of the etched Ni surfaces and etching time is shown in Figure 8. The trends in the SCAs are similar when etched by the same acid. From Figure 8, it can be seen that the SCAs change with the etching time, as the etching time increase, the SCAs on the surfaces increase quickly. After an etching time of 42 h, the SCAs on the surfaces etched by HCl (10.2 wt%, 15.1 wt%, and 20.0 wt%) reach the maximum of 144.5°, 143.3°, and 144.4°, respectively. Thereafter the SCAs gradually decrease (Figure 8a). The trend in the SCAs on the surface etched by HF is similar to the one etched by HCl. The SCA increases rapidly and reaches the maximum of 136.6° in 28 h and then decreases continually (Figure 8e). Similar behavior is observed in samples etched by H_2_SO_4_, H_3_PO_4,_ and HNO_3_, there are two peak values of the SCAs with different optimizing etching time, as shown in Figure 8b–d.

The observed differences in SCAs are due to the different surface roughness, result from the surface morphologies achieved with the different etching times, as shown in Table 1 and Figure 7. The larger values of SCAs depend on the microstructures with larger surface roughness obtained by the etching treatments after a special etching time. When the etching time exceeds the critical value, the building rough structures are destroyed by excessive etching, the surface roughness decreases, as a result, the SCAs decrease. The results suggest that controlling the proper etching times in order to obtain favorable morphologies is the key factor in obtaining hydrophobicity.

In order to study the effect of etchants on the hydrophobicity of etched Ni surfaces, the etchants with different acid compositions are employed. Figure 8 shows the variations in the SCAs of Ni samples etched by different acidic solutions as a function of etching time. The SCAs rapidly increase and the surfaces exhibit hydrophobicity with the SCAs larger than 120.0° after etching by different acid compositions. Figure 8 clearly illustrates that the SCAs on the surface etching by the same acid with different concentration change slightly. However, the SCAs are different when etched by different acidic solutions. After etching by 20.0 wt% HCl, 20.0 wt% H_2_SO_4_ and 40.0 wt% HF for 7 h, the SCAs reach above 110°. The nickel surfaces can obtain the best hydrophobicity with the SCAs bigger than 140.0° after etching by the HCl, and H_2_SO_4_. While the maximum values of SCA on the surface etched by HF and HNO_3_ are 136.6° and 125.1°, respectively. The microstructures with larger surface roughness achieved by the etching treatments using HCl and H_2_SO_4_ are responsible for the larger values of SCAs. Consequently, choosing the proper etchant is favorable to obtain suitable surface morphologies for the hydrophobicity.

## 4. Conclusions

The hydrophobic surfaces on pure nickel substrates were successfully prepared via a one-step chemical etching process without introducing low surface free energy materials. The SCA on the Ni surface was measured to determine its wettability, and the etched Ni surfaces exhibited high hydrophobic properties with the SCA higher than 125°. The results of surface chemical compositions confirm that the Ni surface was almost not changed, there were almost no polar moieties on the surface after chemical etching, except part of the surface was oxidized. The nickel surfaces were packed with various and irregular rough microstructures. According to the analysis of the surface chemical composition and morphology, the nickel surfaces with the presence of terrace-/crater-/thorn-like rough microstructure contributed to excellent hydrophobicity. After optimizing the etching time and etchants, the best hydrophobicity on Ni surface was produced with the SCAs as high as 140.0°. The results demonstrate that it is possible to construct hydrophobic surfaces on hydrophilic substrates by tailoring the surface microstructure using a simple chemical etching process without any further hydrophobic modifications by low surface energy materials, and the process could easily extend to other metal materials.

## 5. Patents

There are four patents (CN201710220397.9, CN201710220626.7, CN201710220627.1 and CN 201710220630.3) resulting from the work reported in this manuscript.

## Figures and Tables

**Figure 1 materials-12-03546-f001:**
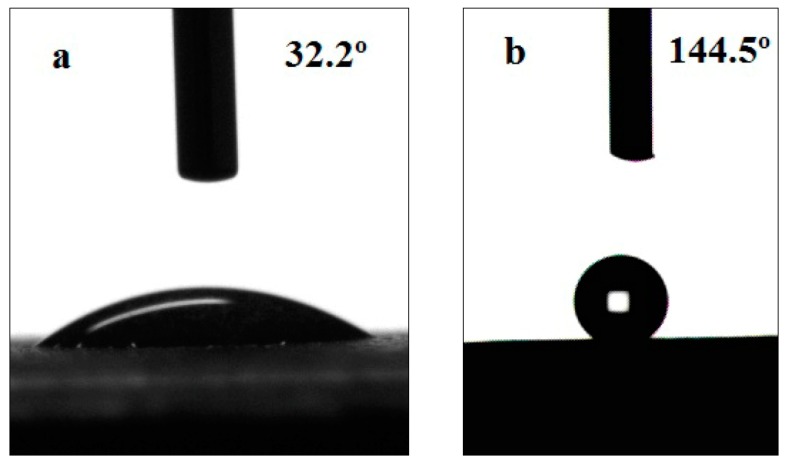
Optical images of water droplets on the surfaces: (**a**) untreated; (**b**) etched by10.2 wt% HCl for 42 h; (**c**) etched by 10.1 wt% H_2_SO_4_ for 42 h; (**d**) etched by 10.0 wt% H_3_PO_4_ for 56 h; (**e**) etched by 10.1 wt% HNO_3_ for 56 h; and (**f**) etched by 40.0 wt% HF for 28 h.

**Figure 2 materials-12-03546-f002:**
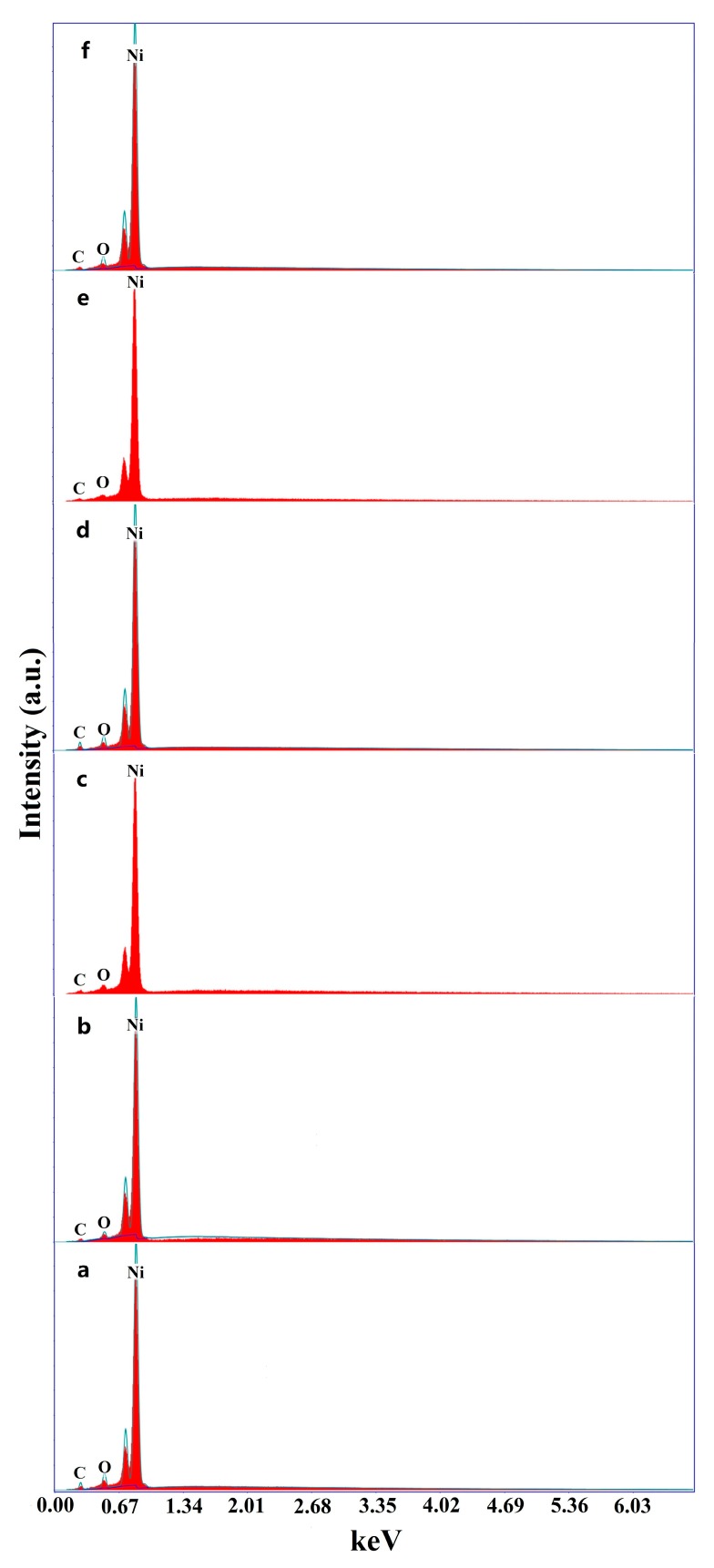
EDS spectra of the Ni surfaces: (**a**) untreated; (**b**) etched by HCl; (**c**) etched by H_2_SO_4_; (**d**) etched by H_3_PO_4_; (**e**) etched by HNO_3_; and (**f**) etched by HF.

**Figure 3 materials-12-03546-f003:**
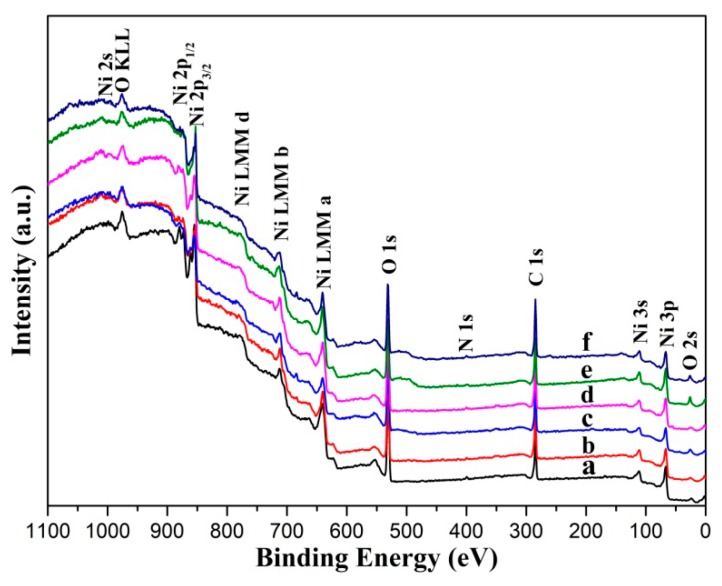
XPS survey spectra of the Ni surfaces: (**a**) untreated; (**b**) etched by HCl; (**c**) etched by H_2_SO_4_; (**d**) etched by H_3_PO_4_; (**e**) etched by HNO_3_; and (**f**) etched by HF.

**Figure 4 materials-12-03546-f004:**
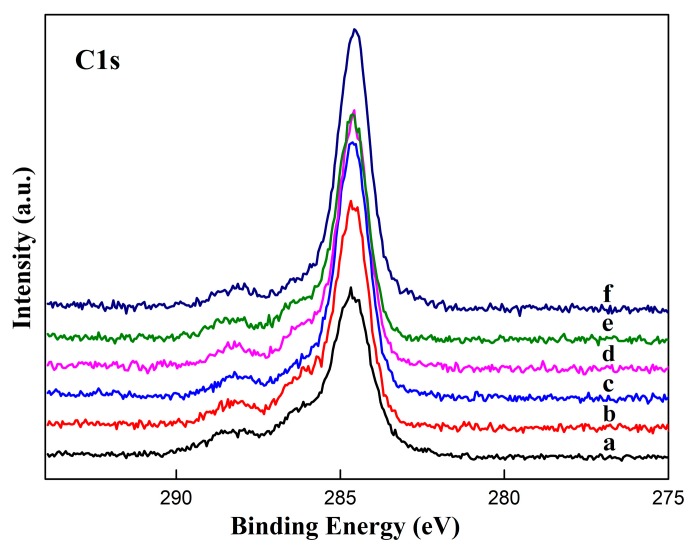
C1s narrow scans XPS spectra of Ni surfaces: (**a**) untreated; (**b**) etched by HCl; (**c**) etched by H_2_SO_4_; (**d**) etched by H_3_PO_4_; (**e**) etched by HNO_3_; and (**f**) etched by HF.

**Figure 5 materials-12-03546-f005:**
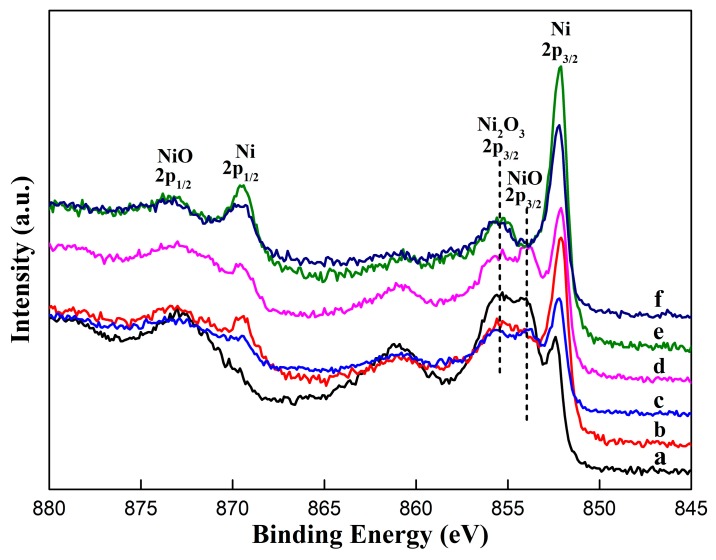
Ni 2p XPS spectra of Ni surfaces: (**a**) untreated; (**b**) etched by HCl; (**c**) etched by H_2_SO_4_; (**d**) etched by H_3_PO_4_; (**e**) etched by HNO_3_; and (**f**) etched by HF.

**Figure 6 materials-12-03546-f006:**
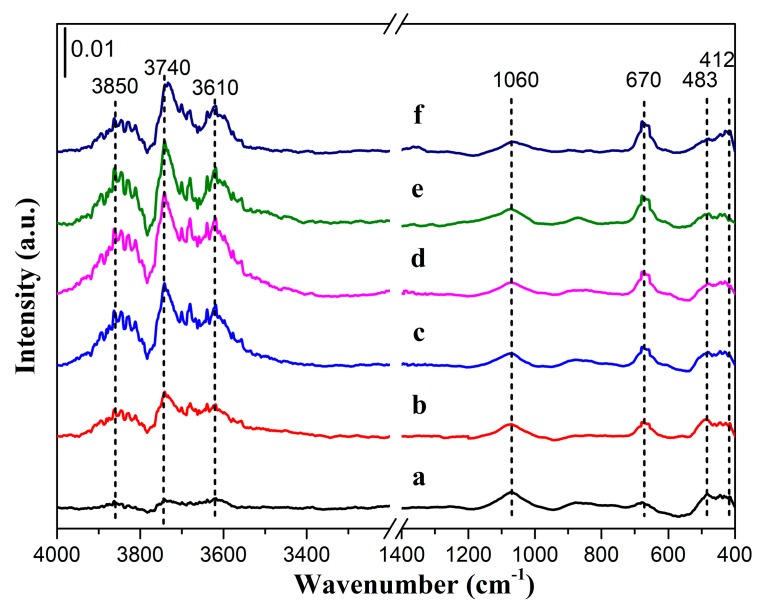
ATR-FTIR spectra of the Ni surfaces: (**a**) untreated; (**b**) etched by HCl; (**c**) etched by H_2_SO_4_; (**d**) etched by H_3_PO_4_; (**e**) etched by HNO_3_; and (**f**) etched by HF.

**Figure 7 materials-12-03546-f007:**
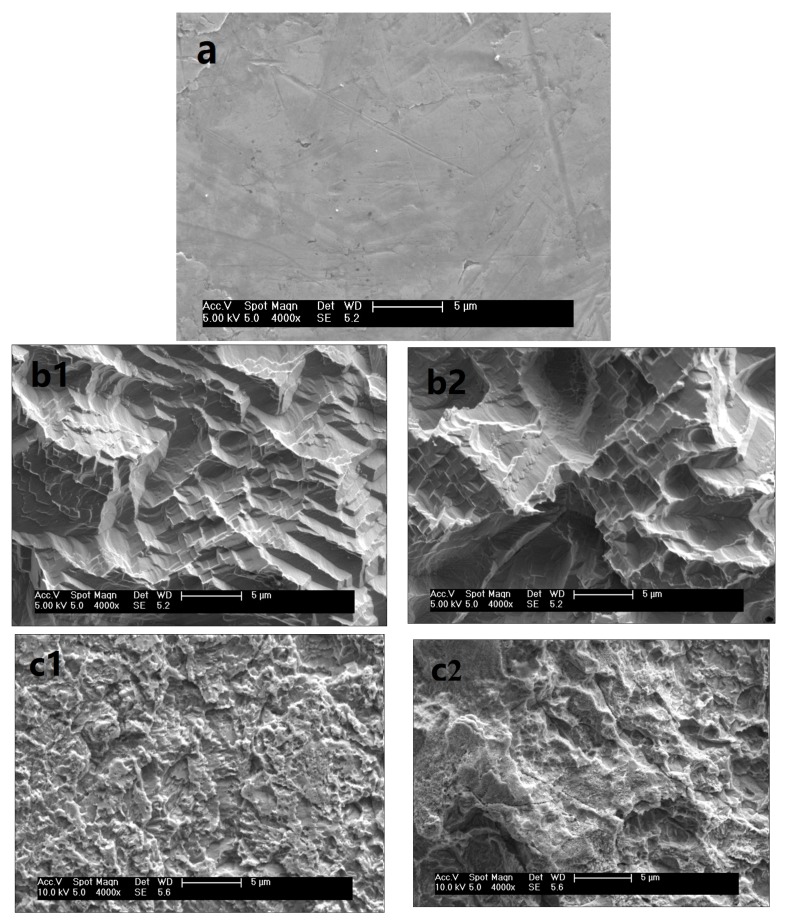
SEM images of Ni surfaces: (**a**) untreated; (**b**) etched by 10.2 wt% HCl for 42 h and 56 h; (**c**) etched by 10.1 wt% H_2_SO_4_ for 42 h and 49 h; (**d**) etched by 10.0 wt% H_3_PO_4_ for 56 h and 63 h; (**e**) etched by 10.1 wt% HNO_3_ for 56 h and 63 h; and (**f**) etched by 40.0 wt% HF for 28 h and 35 h.

**Figure 8 materials-12-03546-f008:**
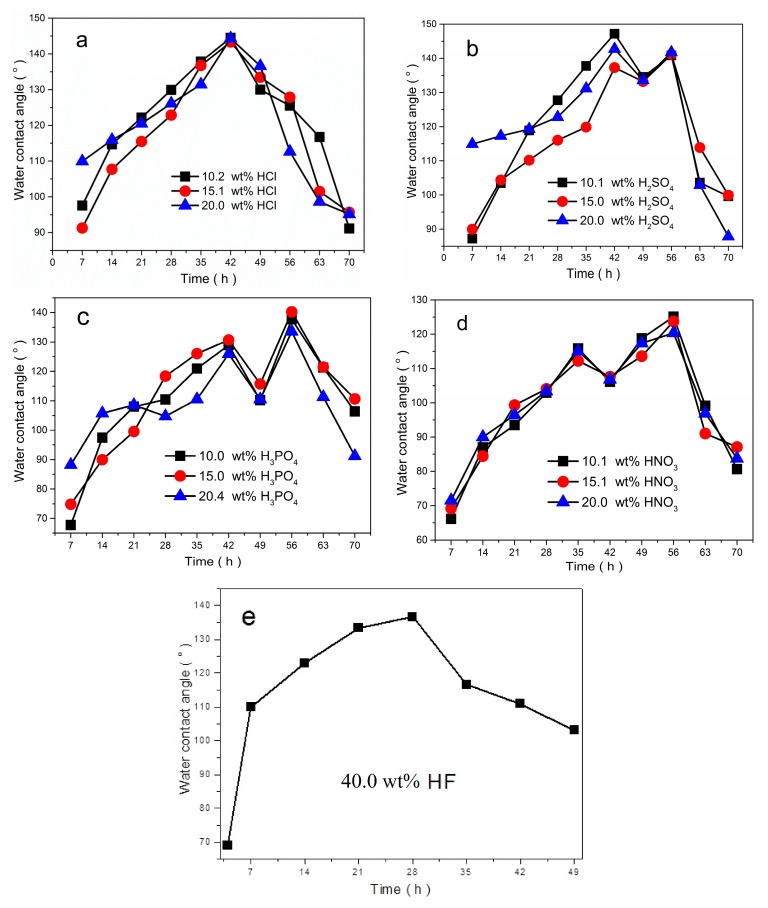
Variations in the SCAs on the etched Ni surfaces with the etching time: (**a**) etched by HCl; (**b**) etched by H_2_SO_4_; (**c**) etched by H_3_PO_4_; (**d**) etched by HNO_3_; and (**e**) etched by HF.

**Table 1 materials-12-03546-t001:** The static water contact angle and surface roughness of Ni metal.

Sample	SCA	Roughness (μm)	SCA	Roughness (μm)
untreated	32.2°	1.33		
10.2 wt% HCl	144.5°	6.24	125.5°	4.58
10.1 wt% H_2_SO_4_	147.2°	6.59	135.5°	5.70
10.0 wt% H_3_PO_4_	137.7°	5.83	121.3°	4.25
10.1 wt% HNO_3_	125.1°	4.58	99.1°	3.29
40.0 wt% HF	136.6°	5.75	116.6°	3.86

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
