# Peer review of "Fabrication of Hydrophobic Ni Surface by Chemical Etching"

_materials, 2019, doi:10.3390/ma12213546_

Round 1

Reviewer 1 Report

The reviewed manuscript describes the hydrophobic Ni surface by chemical etching. The authors present a simple method for increasing hydrophobicity on Ni surface using various acid. The results are interesting and reasonable, but there are some issues to address.

At first, in SEM images, the scale of control image was different from other images. Authors should correct to same scale. Reviewer agree the roughness affects on the hydrophobicity, but there is no value of roughness from SEM image or other data in manuscript. Authors should measure the roughness change as etching time goes on. In addition, NiO can be formed after etching of acid. However, many works has been already published about superhydrophobic surface of NiO. To address it, authors should present their own novelty of this work. 

Reviewer 2 Report

In this manuscript, the fabrication of hydrophobic Ni Surface is presented by using chemical etching method. The manuscript is well written and provide some novelties in this field. In addition, there is a wide experimental work in order to characterize the surface wettability. It is opinion of this reviewer that the manuscript can be accepted for publication in the current form without any further revision.

Reviewer 3 Report

This paper studies the effect of the diverse chemical etching dissolution on the wettabillity of the pure nickel. The surface processing of the samples were conducted via traditional chemical etching methods but new for wettabillity applications. Samples wettability and its evolution with the time  were adequately analysed in this study. The analyses of the chemical composition and topography of the chemical etched samples were correctly carried out to determine the influence of each factor in the wettability of the samples.

This document reaches its main objective but it is recommendable to consider the following comments regarding the paper: 

Roughness of the samples before and after chemical etching could be included in the part 2.1. Materials.

Etchants' concentrations in the etching solution should be include in part 2.2 Samples preparation

The immersion time of the samples in etching dissolution could be specified in part 2.2. Samples preparation.

The type of water used for contact angle measurements e.g. distilled water, fresh water or desionised water, should be included in part 2.3 characterisation.

My specific comments can be found below:

To replace o by ° in:

11 line “125o

17 line “140.0o

42 line “111.5o

43 line “118o

45 line “154.8o

80 line “40oC”

110 line “32.2o

229 line “144.5o to 125.5o

231 line “147.2o to 135.5o

262 line “120.0o

266 line “110o

266 line “140.0o

268 line “136.6o and 125.1o

276 line “125o

283 line “140.0o

To define the acronyms from 55 Line; EDS, XPS, ATR-FTIR and FE-SEM.

To include references in:

109-110 lines "Nickel and its alloys are a type of hydrophilic material with a SCA of 32.2o (REF)"

128 line "oxygen can be attributed to eh background of the EDAX instrument (REF)"

141 line "nitrogen elements in the air were immediately adsorbed into surface (REF)"

163 line "NiO are located at about 854eV span a 0.3eV (REF) range in Figure 5a, c and d."

164 line "Ni2O3 are within about 0.2eV of 855.6eV (REF) for all samples."

199 line "wettability (REF)"

214 line "the water droplet, which can prevent the liquids contacting the nickel substrate (REF)."

To define the acronym from 128 Line; EDAX.

Reviewer 4 Report

Experiments of the Ni surface treatment were well prepared, and the results are reliable. However, the aim of the work it is not novel and a lot of paper presents the results about etching of Ni. 

Introduction: aim of the work and novelty should be better described

Fig. 1.- smaller images of the drops should be prepared to save the place in manuscript

Fig.2. it is not significant information on the EDX spectra, besides Ni, C, O which is obvious, maybe the results might be presented in one figure

results and Discussion- no discussion is visible in the manuscript, the discussion part must be improved.

Round 2

Reviewer 1 Report

The authors has revised the manuscript followed by reviewer's comment. The manuscript has become well-organized.